# Laser Cutting of Non-Woven Fabric Using UV Nanosecond Pulsed Laser

**DOI:** 10.3390/mi15111390

**Published:** 2024-11-17

**Authors:** Jiajun Fu, Chao Liu, Runhan Zhao, Huixin Wang, Zhongjie Yu, Qinghua Wang

**Affiliations:** 1School of Mechanical Engineering, Southeast University, Nanjing 211189, China; 220230419@seu.edu.cn (J.F.); 220220348@seu.edu.cn (C.L.); a13292804086@126.com (R.Z.); yzj2584314459@163.com (Z.Y.); 2Jiangsu Key Laboratory for Design and Manufacture of Micro-Nano Biomedical Instruments, Nanjing 211189, China; 3Institute of Agricultural Facilities and Equipment, Jiangsu Academy of Agricultural Sciences, Nanjing 210014, China; wanghuixin@jaas.ac.cn; 4Key Laboratory of Protected Agriculture Engineering in the Middle and Lower Reaches of Yangtze River, Ministry of Agriculture and Rural Affairs, Nanjing 210014, China

**Keywords:** non-woven fabric, nanosecond laser cutting, surface morphology and chemistry, carbonization

## Abstract

The efficient cutting of non-woven fabric shows great significance to the development of the textile industry. In recent years, laser cutting technology has been widely applied in the clothing industry due to its high efficiency and cutting quality. In this work, a UV nanosecond pulsed laser with a wavelength of 355 nm and a max power of 6.5 W is used to cut non-woven fabric with a thickness of 0.15 mm. The variation of kerf width, surface morphology, and chemical contents are investigated under different laser processing parameters, and the optimal processing parameter is determined. The experimental results demonstrate that the degree of crystallization and chemical composition of the kerf on the non-woven fabric surface is significantly influenced by laser cutting parameters such as laser scanning speed (from 100 to 700 mm/s) and frequency (from 20 to 70 kHz). The scanning speed of 500 mm/s and frequency of 30 kHz are considered the best parameters for achieving abundant energy for the complete and efficient cutting of non-woven fabric. In addition, the level of carbonization and oxidation reaches a relatively low value, and the kerf width is 0.214 mm, which is considered a reasonable value under the optimal processing parameters, showing high cutting quality. Furthermore, the effect of different cutting treatments on surface morphology and chemical contents is also studied. The experimental results present that the non-woven fabric cut by laser possesses a flat kerf, showing a similar effect to that of scissor cutting. Moreover, due to the programmability of laser processing patterns, it is possible to create more intricate designs on non-woven fabric. This facilitates the application and promotion of laser-cut non-woven fabrics. These results can provide a certain reference for laser cutting in the textile industry and are expected to allow for the cutting of high-quality kerf with low carbonization and oxidation.

## 1. Introduction

Fibrous fabric is widely used in medicine [1,2], clothing [3], industry [4], and other fields due to its characteristics of moisture resistance, breathability, flexibility, flame retardancy, low price, recyclability [5,6,7,8,9], etc. For non-woven fabric, the traditional methods of cutting mostly use knives, which causes easy wear of materials [10]. In addition, cutting different patterns and shapes requires diverse tools, which can easily bring about low efficiency in cutting work [11]. Therefore, it is of great significance to seek a simple and efficient method for cutting non-woven fabric.

In addition to laser cutting of hard materials such as metals and ceramics, some studies have also researched the feasibility of laser cutting methods for soft and stretchable materials such as silk and cotton. Hung et al. [12] obtained cotton-knitted single fabrics treated with different laser processing parameters. The experimental results demonstrated that small grooves and cracks appeared on the fiber surface after laser ablation and appropriate laser parameters can improve the breaking strength of the fabric. Liou et al. [13] used a CO_2_ laser to cut carbon fiber fabric and discussed the effects of different combinations of processing parameters. The experimental results presented that some deep holes appeared on the carbon fiber surface, leading to the phenomenon that the surface became rough and looked darker than the untreated surface. In addition, CO_2_ laser ablation could improve its aesthetic performance by changing its light reflection behavior. Dai et al. [14] obtained five kinds of thermal insulation layers (the outer layer (Nomex^®^IIIA), moisture barrier (I-70/ PTFE: polytetrafluoroethylene), thermal barrier (Nomex felt), and comfort layer (flame-resistant viscose)) through laser cutting technology and the multilayer compound method. The experimental results showed that the cross-section structures obviously affect the thermal protection performance, and the a-type structure exhibits a lower moisture resistance and higher total heat loss compared to the i-type structure. Chan et al. [15] used a CO_2_ laser with different laser processing parameters to process cotton and cotton/polyester blended fabrics, and then the surface structures of the sample were analyzed. The experimental results indicated that the density and size of pores and cracks in cotton material increased with the laser intensity. Moreover, the measurement results of fabric weight and thickness confirm that laser treatment can change the weight of cotton and cotton/polyester blended fabrics, and the fabric weight of cotton and cotton-polyester blended fabrics decreased with the increase of laser processing parameters. Zoghi et al. [16] developed a theoretical model for estimating the kerf depth based on the material properties and laser parameters of blended fabrics. The effects of laser processing parameters such as power and scanning speed on the kerf depth and speed were investigated by using a CO_2_ laser to cut fabric. The experimental results showed that the combination of laser parameters with an output power of 45 W and a speed of 30 mm/s can cut 10 layers of fabric while maintaining excellent quality. Although these methods have proved the feasibility of cutting fabric using lasers and discussed the physical and chemical variation of the fabric, the large heat-affected zone of the CO_2_ laser leads to a bigger kerf, and a high degree of crystallization of the kerf can easily result in a rough feel and can even scratch the skin [17]. Furthermore, the operation is cumbersome while processing fine patterns and tends to damage fabric with low thicknesses. Hence, it is of great necessity to develop a better method to improve laser cutting for non-woven fabric.

In this study, a UV nanosecond pulsed laser with a wavelength of 355 nm is used to cut non-woven fabric. A “fracture” structure was used to measure the cut quality obtained on the non-woven fabric surface under the thermal action of laser cutting. A thin and narrow kerf can be obtained under low scanning speed and frequency because it is conducive to the energy absorption of non-woven fabric. In addition, the surface chemical element composition is also characterized to analyze the chemical variation. The experimental results show that the degree of crystallization and chemical composition of the kerf is significantly influenced by laser cutting parameters such as laser scanning speed and laser frequency. Furthermore, the optimal laser cutting parameters are confirmed by the kerf width degree of crystallization, which can provide a certain reference for laser cutting in the textile industry.

## 2. Materials and Methods

### 2.1. Materials

The type of non-woven fabric used in this experiment is hot-air non-woven fabric purchased from Qidong Hean Non-woven Material Co., Ltd. (Qidong, Jiangsu, China). The color is white, and the main component is polyester fiber (PET). Moreover, the weight of the non-woven fabric is 40 g/m^2^ and the thickness is 0.15 mm. In addition, the non-woven fabric possesses the properties of nontoxicity, and the thermal conductivity is 0.025~0.035 W/(m·K).

### 2.2. Laser Cutting Experiments

Figure 1 demonstrates the laser cutting process of non-woven fabric. The non-woven fabric is fixed on the sample table and forms a “fracture” and carbonized zone at the kerf. The processing pattern is designed by CAD 2020 software on a computer, and then the laser cutting parameters are set. Before the experiment, the position of the focal plane is determined by finding the position of the minimum light spot on the stage, and the focal plane position is kept unchanged. The processing instructions are issued to the controller during processing, and a laser beam (Seal-355-3/5, Shenzhen) is emitted. After that, the laser beam passes through the mirror, attenuator, and beam expander and finally enters the scanning galvanometer (MQ5T, Guangzhou, https://www.cf388.com/product-category/portable-laser (accessed on 2 August 2024)). The non-woven fabric is cut by the laser beam with different processing parameters, as shown in Table 1.

### 2.3. Surface Characterizations

A metallographic microscope (BX53M, 50×) and digital microscope (Leyue T16, 50×) (from Mumuxili, Nanjing, China) are used to characterize the surface morphology of the kerf. A high-resolution sputtering coater (Q150TS) is utilized to “spray gold” to enhance the conductivity of the non-woven fabric. After the sample is sprayed with platinum, a field emission scanning electron microscopy (SEM, Hitachi Regulus 8100) is used to characterize the surface morphology of the kerf on the non-woven fabric. The acceleration voltage is set to be 15~20 kV. SEM images with different magnifications are selected for analysis and comparison. The chemical element composition of the kerf under different processing parameters is measured by energy-dispersive X-ray spectroscopy (EDS, Sirion) under an ultra-vacuum environment and X-ray photoelectron spectroscopy (XPS) to observe the chemical changes caused by laser cutting.

## 3. Results and Discussions

### 3.1. Width of Kerf

It is known that laser processing parameters play an important role in cutting effect and quality [18]. The relationship between the laser frequency and the power, pulse width, and power density can be found in Table 1. It can be seen that the single pulse energy and power density both increase first and then decrease with the increase of frequency and reach the peak value at 30 kHz. Subsequently, the influence of laser processing parameters on the kerf width is investigated. As shown in Figure 2, the kerf width decreases with the increase of scanning speed. Moreover, the kerf width shows a trend of increasing first and then decreasing with the increase of the frequency and finally reaches a peak value of 1.278 mm at a scanning speed of 100 mm/s and a frequency of 30 kHz. This is because the increase in scanning speed leads to the gradual number reduction of laser pulses applied to the single irradiation point, resulting in less energy applied to the fabric while the single pulse energy remains constant [19]. Furthermore, the number of single laser pulses increases with the increase in frequency, but the energy and power density of the single laser pulse decrease. In this case, the short duration causes the direct breakage of the chemical bond inside the non-woven fabric, forming a preliminary fracture [20]. As the energy is further absorbed, a burning zone appears around the kerf and splashes out the ionized and vaporized material, leading to the kerf width first increasing and then decreasing [21]. While the scanning speed is higher than 600 mm/s, the instantaneous thermal effect is unable to fuse the fabric leading to a kerf width of 0, meaning that the non-woven fabric is not cut off. The change of the kerf width corresponds well with the change of the pulse energy, as shown in Table 1. Obviously, the energy absorbed by the non-woven fabric is higher while the single pulse energy is high, which will induce a bigger kerf width.

### 3.2. Effect of Scanning Speed on Surface Morphology and Chemical Composition of Non-Woven Fabric

To explore the effect of laser scanning speed on the surface morphology of non-woven fabric, four representative images at different scanning speeds with frequency controlled at 40 kHz are obtained (Figure 3). It can be seen from the digital micrographs that the kerf width decreases with the increase in scanning speed. While the scanning speed is higher than 600 mm/s, the non-woven fabric is not completely cut off. By comparing the SEM images (200× and 500×) of the kerf of non-woven fabric at different scanning speeds, it can be found that the scanning speed has a significant impact on the kerf structure. Figure 3a displays that as the scanning speed is 400 mm/s, long and thin carbonized pellets appear at the margin of the kerf because the laser processing time of the single point and the energy absorbed is increased in the reaction region, resulting in the increase of the number of fuses [22]. In the meantime, the spherical structure formed by melting evaporation is more complex, and the density and surface carbonization degree increase significantly, which eventually contributes to a larger kerf width [23]. As shown in Figure 3c,d, some spinning and winding connections still exist in the middle of the fracture, and the non-woven fabric is not completely cut off. This is mainly because the high scanning speed results in a decrease in the melting quality of the non-woven fabric [24]. Figure 3b exhibits the images of the non-woven fabric when the scanning speed is controlled at 500 mm/s. It can be seen that the degree of carbonization and density of the spherical structure is moderate. Moreover, the kerf on the surface is orderly and leads to a smooth touch, showing great significance in realizing the efficient cutting of non-woven fabric.

Apart from surface morphology, the proportion of chemical element content in the kerf of non-woven fabric at different laser scanning speeds is experimentally determined (Figure 4). Slight data fluctuations can be seen as the normal error margin in the measurement. Four representative samples processed using different scanning speeds are selected to conduct the XPS analysis. Before the analysis, all spectra were charge-shifted to confirm the accuracy of the data. As shown in Figure 4a, the main elements with high energy density are mainly C and O, and the binding energies of C 1s and O 1s are 284.8 and 532.1 eV. Meanwhile, the elemental contents of C and O are measured, as demonstrated in Figure 4b. It is presented that scissor cutting has no effect on the change of chemical element content. With the increase of scanning speed, the proportion of element carbon (C) decreases first and then increases and reaches the least value (84.64%) at the scanning speed of 500 mm/s while the content proportion of element oxygen (O) reaches the peak value (15.36%). This is mainly attributed to the thermal decomposition reactions, which can be divided into two stages, and oxidation reactions, which occur when the non-woven fabric is cut by laser [25,26,27,28]. The first stage is the thermal decomposition reactions of the main chain of polyester macromolecules, and the products are mainly carboxylic acid and vinyl ester. At this stage, the C-C(H) bonds in the main chain macromolecules break and combine with oxygen to form C-O bonds, ether bonds, and O-C=O, which corresponds to the downward trend of the content of C-C(H), as shown in Figure 4c–f. With the further increase of the energy absorbed by the non-woven fabric, the second stage is the further decomposition of the benzene and the unstable initial carbon layer, and the products are mainly CO_2_ and H_2_. Besides, the intensity of C1’s peaks shows obvious variation. The content of C-O reached a maximum value (22.25%) at the scanning speed of 700 mm/s compared with the speeds of 300 and 500 mm/s. This is mainly due to the energy absorption of the C-O bond formed in the first stage: the C-O breaks to reform free C and O and recombines to form new products such as terephthalic acid [27]. The recoil pressure generated during the process of gas escape makes the edge of the non-woven fabric curl inward. Under the condition that the scanning speed is higher than 500 mm/s, the energy absorbed by the fabric is insufficient for the reactions of the second stage reaction, resulting in the decreased degree of the thermal decomposition reaction and the reduction of the amount of generated gas. The absorbed energy of the fabric is enough while the scanning speed is lower than 500 mm/s, leading to a decreased decomposition degree of the initial carbon layer and contributing to the increase of C atom content and almost no change in O atom content [29].

### 3.3. Effect of Frequency on Surface Morphology and Chemical Composition of Non-Woven Fabric

Figure 5 illustrates the surface morphology of the kerf of non-woven fabric at different frequencies when the scanning speed is controlled at 500 mm/s. The surface structure of the kerf of non-woven fabric is significantly affected by the frequency. It can be seen from the digital micrograph that the kerf width increases and then declines with the increase of frequency, and the non-woven fabric is not completely cut off when the frequency is set at 50 kHz. Under the condition that the frequency is 20 kHz, the SEM images demonstrate that the number of fusing and the degree of carbonization are greatly reduced, and the density of the spherical structure formed by melting evaporation is obviously cut down. There are still a few entanglements of fiber lines around the kerf. Moreover, the kerf width at 20 kHz also decreases compared with that of the frequency of 30 kHz. As for the frequency at 40 kHz, the kerf width decreases again, and the carbonized pellets are smaller and sparser. This can be attributed to the fact that the laser energy per unit time incident on the unit area of the non-woven fabric material is reduced [30]. Therefore, the temperature of the carbonization layer is low when the material is laser ablated, and the energy transferred from the carbonization layer to the thermal decomposition layer is reduced [31]. In this case, the speed of the thermal decomposition reaction of the material is smaller, resulting in a slower mass ablation rate of the material, and a narrower kerf width is formed. Moreover, it can be seen in Figure 2 that the difference in kerf width among these three scanning frequencies is relatively small. When the scanning frequency is 30 kHz, the single pulse energy and energy density of the laser are the highest. Therefore, considering the uneven distribution of fibers in non-woven fabric during the actual processing, a parameter combination with higher cutting ability is more applicable.

In addition, the proportion of chemical element content in the kerf of non-woven fabric, as shown in Figure 6, is measured for XPS analysis. As shown in Figure 6a, similar to Figure 4a, the main elements of non-woven fabric at different frequencies are C and O. It can be seen from Figure 6b that with the increase of frequency, the proportion of element C fluctuates and reaches the peak value (85.17%), while the proportion of element O reaches the least value (14.83%) at the frequency of 30 kHz. In the meantime, the content of C-C at 20 kHz and 60 kHz exhibits a significant decline compared with that of the untreated non-woven fabric. This is mainly related to the pulse energy absorbed at a single point of the non-woven fabric. The single pulse energy and power density reach the peak value at the frequency of 30 kHz, and thus the thermal decomposition reaction mainly occurs at the first stage resulting in an increase in the decomposition degree of the terephthalic acid [32]. The high decomposition degree of the terephthalic acid leads to an increase in the amount of generated CO_2_ gas. At this time, the content of C and O decreases, and the change in the proportion of O content becomes more distinct, which shows that the proportion of C content increases while the proportion of O content decreases compared to that of 20 kHz. While the frequency further rises to 40 kHz, the energy absorbed by the fabric is reduced, leading to a further reduction of the degree of oxidation reaction and thermal decomposition reaction. The number of vinyl ester and terephthalic acid produced by the decomposition of the main chain of polyester macromolecule is decreased, contributing to the content change as shown in Figure 6. The experimental result further exhibits the importance of reasonable control of laser processing parameters for the efficient cutting of non-woven fabric.

### 3.4. Surface Morphology and Chemical Contents Affected by Different Treatments

To verify the effectiveness of the laser cutting technique developed in this work, the surface morphology and chemical contents obtained by scissor cutting and laser cutting are measured for analysis and comparison. As demonstrated in Figure 7(a1–a3), the surface of the untreated non-woven fabric consists of lines intertwined with each other, and there is no fracture. Figure 7(b1–b3) shows the characterization results of the surface morphology for the non-woven fabric cut by scissors. It can be seen that an obvious kerf appears on the non-woven fabric, which has been completely cut off under the effect of shear force. The cutting section is uneven, and the surface roughness is low. In addition, the fibers of the non-woven fabric show high flatness after being cut by the scissors, and there is no spherical structure formed by melting evaporation on the kerf. When the non-woven fabric is cut by laser, it can be observed that the kerf appears flatter, and the kerf width is relatively narrow. Furthermore, there are some carbonized pellets generated on the kerf of the non-woven fabric due to the thermal effect of laser ablation (Figure 7(c1–c3)), and reducing the degree of carbonization is the key to improving the flatness and feeling of the kerf [33,34].

The different cutting principles of scissor cutting and laser cutting are investigated through the chemical elemental analysis results. Figure 7(a4) shows that a large amount of C (96.88%) and slight O (1.92%) elements can be detected on the surface, mainly owing to the fact that the non-woven fabric principally consists of PET. There is a small amount of platinum (Pt) elements detected on the surface, which is attributed to the spraying of Pt film on the surface before SEM tests. Moreover, the ratio of C and O element contents of the untreated non-woven fabric is higher than that of PET, which is due to the slight pollution of non-woven fabric exposed to air and the carbon-containing compounds remaining in the equipment during the detection [35]. As can be seen in Figure 7(b4), the measurement result of chemical element content on the surface of the non-woven fabric indicates that scissor cutting has no influence on the variation of the chemical element content. Figure 7(c4) presents the EDS result on the non-woven fabric surface treated by laser cutting. It is demonstrated that the content of the O element increases significantly to 3.07%, while the content of the C element decreases slightly. This is mainly due to the thermal effect of laser cutting, which results in a strong oxidation reaction of the non-woven fabric at the kerf, leading to the increase of the content of the O element [36]. The main reason for this element variation is the different reactions that occur under the two cutting methods. When cutting with scissors, the non-woven fabric experiences shear deformation and fracture under the action of force [37,38], and the internal structure of the non-woven fabric is not destroyed. As shown in Figure 7(b3), it can be seen that obvious flat fiber bundles appear around the kerf. On the contrary, the main chain of macromolecules inside the PET fiber is broken under the action of the high-energy laser beam. Moreover, the non-woven fabric absorbs heat so that the temperature rises rapidly during laser cutting. The material begins to vaporize and quickly escape within a short time duration, forming a preliminary incision in the fiber break [39]. The fiber bundles at the fracture further absorb heat, begin to coke, and slowly curl and contract to form carbonized pellets, as shown in Figure 7c [40].

## 4. Conclusions

In this work, a UV nanosecond pulsed laser cutting technology is developed to cut non-woven fabric. A “fracture” structure is obtained on the non-woven fabric surface under the thermal action of laser cutting. Subsequently, the influence of laser processing parameters on the kerf of non-woven fabric is investigated, and the surface morphology and chemical composition are characterized. Finally, the difference in surface morphology and chemical composition obtained by scissor cutting and laser cutting is compared. The following conclusions are obtained:The ablative oxidation and carbonization occur on the non-woven fabric surface, resulting in the generation of a carbonized spherical structure around the kerf. In the meantime, the effect of laser scanning speed and frequency on the kerf width and micro-nano structures of non-woven fabric surfaces is systematically investigated.Under the premise that the non-woven fabric can be completely cut off efficiently at one time and the kerf carbonization degree can be reduced to the greatest extent, a narrow kerf can be obtained when the laser processing parameters are set at a scanning speed of 500 mm/s and a frequency of 30 kHz. The results of this experiment are applicable to the non-woven fabric used in this work, and further experiments are needed in the future to improve the applicability of this work.Finally, the non-woven fabric treated by laser cutting has a smooth feeling similar to that of traditional cutting methods such as scissor cutting.

## Figures and Tables

**Figure 1 micromachines-15-01390-f001:**
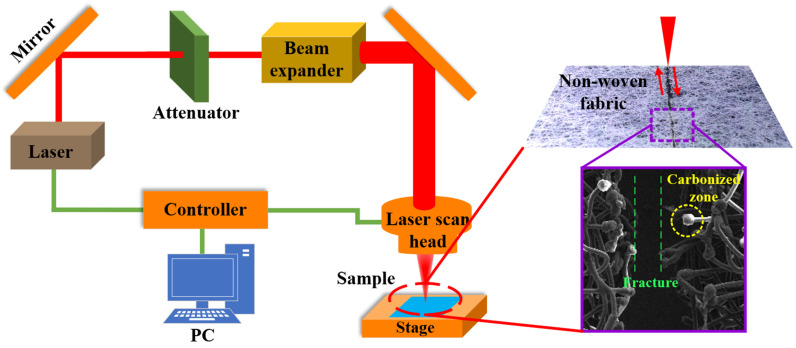
Process schematic for laser cutting of non-woven fabric.

**Figure 2 micromachines-15-01390-f002:**
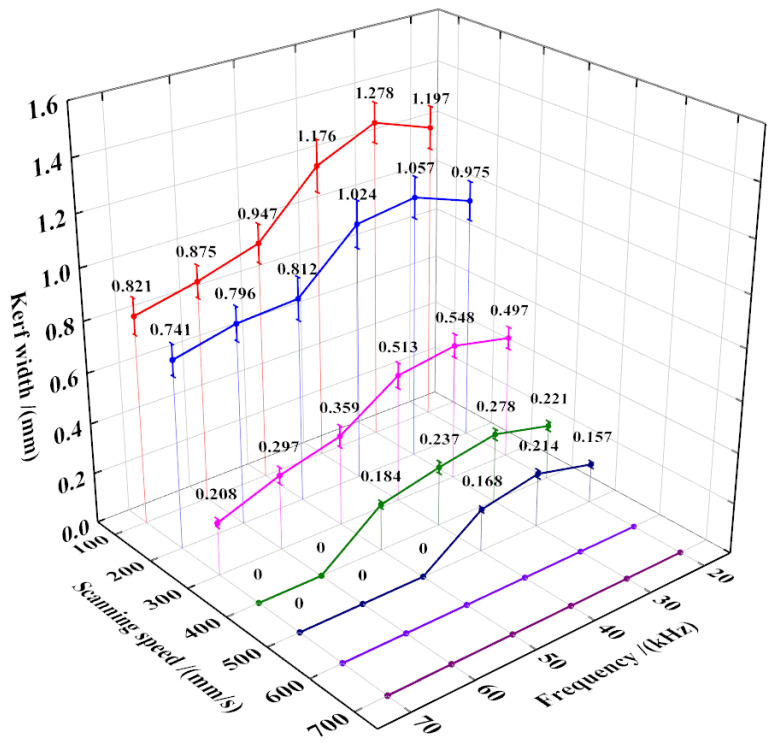
Kerf width of non-woven fabric obtained under different laser processing parameters.

**Figure 3 micromachines-15-01390-f003:**
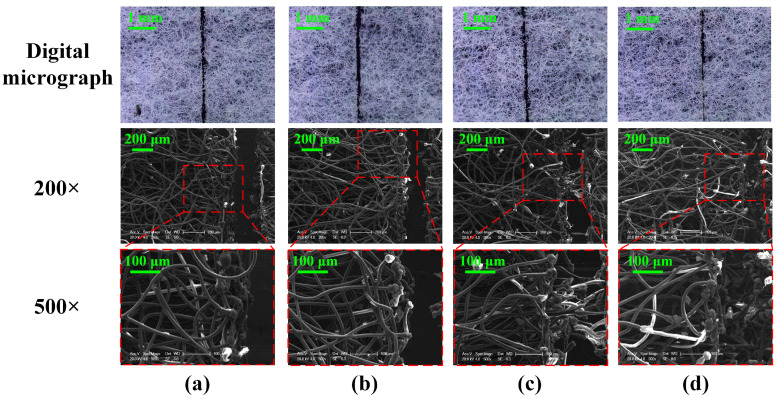
Surface morphology of the kerf of non-woven fabric at different laser scanning speeds: (**a**) 400 mm/s; (**b**) 500 mm/s; (**c**) 600 mm/s; (**d**) 700 mm/s.

**Figure 4 micromachines-15-01390-f004:**
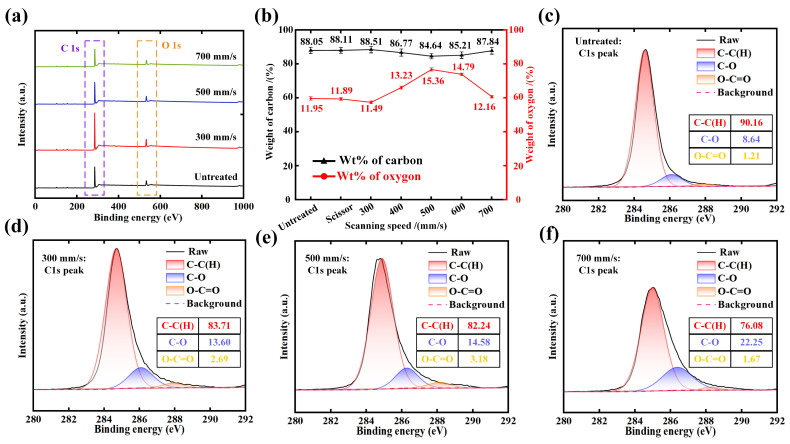
Chemical composition analysis: (**a**) The full spectrum at different scanning speeds; (**b**) The proportion of chemical element content at different speeds; (**c**–**f**) The fine spectrum of C1’s peak under different treatments.

**Figure 5 micromachines-15-01390-f005:**
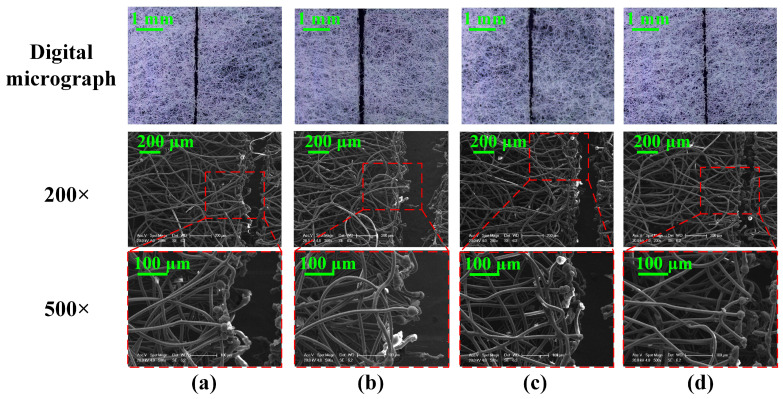
Surface morphology of kerf of non-woven fabric at different frequencies: (**a**) 20 kHz; (**b**) 30 kHz; (**c**) 40 kHz; (**d**) 50 kHz.

**Figure 6 micromachines-15-01390-f006:**
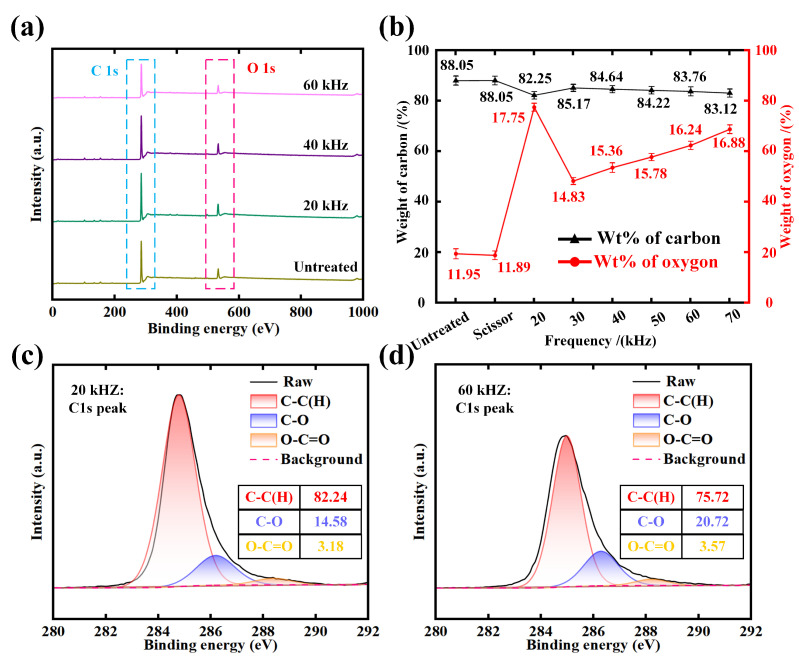
Chemical composition analysis: (**a**) The full spectrum at different frequencies; (**b**) The proportion of chemical element content at different frequencies; (**c**,**d**) Fine spectrum of C1’s peak under different frequencies.

**Figure 7 micromachines-15-01390-f007:**
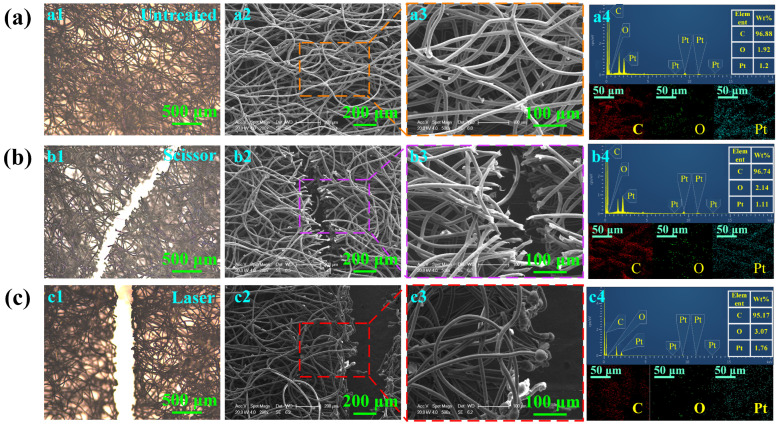
The influence of different treatment methods on the surface morphology and chemical composition of non-woven fabric: (**a**) Untreated; (**b**) Scissor cutting; (**c**) Laser cutting.

**Table 1 micromachines-15-01390-t001:** Laser processing parameters used in this work.

Frequency (kHz)	Average Power (W)	Pulse Width (ns)	Pulse Energy (µJ)	Peak Power (MW)	Power Density (GW/cm^2^)
20	3.53	8.97	176.50	0.0197	0.6959
30	6.09	9.91	203.00	0.0205	0.7245
40	6.7	11.43	167.50	0.0147	0.5183
50	6.37	13.01	127.40	0.0098	0.3463
60	5.82	14.39	97.00	0.0067	0.2384
70	5.21	16.19	74.43	0.0046	0.1626

## Data Availability

All data will be available upon request.

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
