# Peer review of "Laser Cutting of Non-Woven Fabric Using UV Nanosecond Pulsed Laser"

_micromachines, 2024, doi:10.3390/mi15111390_

Round 1
Reviewer 1 Report (New Reviewer)
Comments and Suggestions for Authors
Dear Authors,
The paper explored the process of laser cutting of non-woven fabric using UV nanosecond pulsed laser. Much attention was paid to the composition of the material on the surface after cutting, and a qualitative comparison was made. The paper can be recommended for publication after a number of issues listed below are addressed.
Specific Comments
1. Lines 14-31 indicate that non-woven fabric can be cut with a laser using comparable cutting characteristics, but what effect and advantages laser cutting provides compared to cutting with scissors is not specified. Please clarify.
2. Lines 61-66 are highlighted for some reason.
3. Lines 182-230, formatting is broken, text highlighting is difficult to read. Please correct.
4. The article completely lacks a Discussion section, no justification or evidence is provided as to why the cut of the material surface is formed with different quality. Without this, this article is not a full-fledged scientific study.
Author Response
Please see the attachment.

Reviewer 2 Report (New Reviewer)
Comments and Suggestions for Authors
The manuscript presents the results of optimization of parameters of laser cutting of non-woven fabric, it can be of narrow interest to specialists in laser cutting of fabrics. The presented results are explained in a rather puzzling manner. The technical result is clear from the graphs and parameters in the table, and the explanation requires adjustment. The manuscript requires serious revision.
1. The introduction is rather long. It contains general information about the laser cutting of metals and ceramics, which is not related to the topic of the manuscript. The second paragraph in the introduction section should be removed, lines 43-55.
2. When surveying the literature data, the composition of the fiber from which the samples were made was not indicated in the introduction section, i.e. reference [20], lines 66-70. This needs to be supplemented.
3. The authors write "Hence, it is of great necessity to develop a novel laser cutting method for non-woven fabric.", lines 88-89. The results presented in the article cannot be considered as a "novel method". The work presents the results of optimizing the cutting parameters (laser wavelength, power, speed, etc.). This should be corrected.
4. Section 2.1 "Materials". What filler/dye gave the non-woven fabric its white color? This is important because it affects the properties of the fabric and the laser cutting conditions.
5. There are errors in Table 1. What is "Peak power (MV)"? Did you mean MW? Also, the laser spot size on the sample should be specified to confirm the power density stated in Table 1.
6. The explanation of the dependence of cutting mechanisms on laser exposure modes in Section 3.1 (and further on) is not clear. For instance, lines 151-152. "In this case, the high photon energy and short duration cause the direct breakage of the chemical bond inside the non-woven fabric, forming a preliminary fracture [26]." Photon energy is determined only by the wavelength of the laser radiation and is the same in all experiments. That is, the contribution of the photochemical processes in single-photon absorption is the same. Or are the authors discussing multiphoton absorption?
7. In the experiments presented by the authors, all the features of the fabric cutting process (cut depth, cut width, edge carbonization) are determined mainly by the temperature and heating zone of the fabric, which depends on the pulse repetition rate and the scanning rate. How do the authors evaluate the temperature in the cutting zone?
8. From what area size were the XPS spectra measured? I believe that the measurement area was much larger than the area exposed to laser action. The data provided depend more on the position of the samples in the device than on the composition of the cut surface. It is difficult to rely on these data.
9. Conclusion:
Conclusion 2 is rather trivial and should be removed.
Conclusion 3 is related only to the laser used by the authors and only for this non-woven fabric. This should be clearly stated.
Conclusion 4 "The experimental results can provide reference for the use of laser cutting technology in the clothing industry." This does not follow from the data presented in the manuscript.
The conclusion should be revised.
Author Response
Please see the attachment.

Reviewer 3 Report (New Reviewer)
Comments and Suggestions for Authors
Comment 1: What is the resolution of the metallographic, digital and SEM microscope they used?
Comment 2: The authors claim that figure 3b demonstrates that «the kerf on the surface is orderliness and leads to a smooth touch, showing great significance to realize the efficient cutting of non-woven fabric», and the results are better compared to figure 3a. However, one could argue that this only holds because the authors choose to demonstrate a specific part of the surface morphology where this is correct. Therefore it is unclear if this holds for all the cut area and not only for this specified region.
Comment 3: Moreover, from the comments the authors made for figure 5 it is unclear to me if the results are better for 20, 30 or 40 kHz. The authors claim both at the abstract and the conclusions that « The scanning speed of 500 mm/s and frequency of 30 kHz provide better results since a narrower kerf is developed » however this is not clear at all from the comments they made about figure 5. They should recheck lines 230-244 and make this statement more justified to the reader. Comment 2 also needs an answer for this case.
Comment 4: The authors should improve the overall quality of English throughout the manuscript. For example in line 93 correct the phrase «can be got».
Round 2
Reviewer 1 Report (New Reviewer)
Comments and Suggestions for Authors
Dear Authors,
The explanations of the work are clear. Thank you! The article can be published.
Reviewer 2 Report (New Reviewer)
Comments and Suggestions for Authors
Comments corrected.
The manuscript can be accepted in precent form.
Reviewer 3 Report (New Reviewer)
Comments and Suggestions for Authors
My points were addressed.
This manuscript is a resubmission of an earlier submission. The following is a list of the peer review reports and author responses from that submission.
Round 1
Reviewer 1 Report
Comments and Suggestions for Authors
Review of:
Laser cutting of non-woven fabric using UV nanosecond pulsed laser
General comments: The authors describe to laser cutting process of an unknown material using a laser source in the UV spectrum. The general outline of the paper is decent but needs a lot of improvements before publication.
Introduction:
Line 40: “As a new cutting technology” – The laser cutting technique is over 40 years old and is rather considered as “new”. Please change.
Line 42 – 52 This section of the introduction does not contribute to the presented work since the goal is not to cut steel or silicate. Moreover, the authors should investigate the state of the art concerning laser cutting of their material (class). Therefore, the state of the art should be shown in more detail since there is far more research done in this field.
Materials and Methods:
This section is called “Materials and Methods”. Somehow the authors forgot about the Materials part to describe. There is no information what soever what kind of “non wovon fabric” was investigated, including thickness, thermal conductivity, and all the other important information that needs to be filled in. With this they should start.
Line 101- 102 Table 1: The Spot diameter is the w86 or 2w value? Furthermore, the pulse width is given with 2 decimals. How are the authors being assure that these are as precise as given? Was it measured?
Line 102- 103 Figure 1: To improve the image of the setup, the beam expander should really “expand the beam”. Here is it the same size before and after.
Line 102- 103 Figure 1: The authors stated that a laser scan head was used. Which scan head from which company? What are the specifications for that component. Furthermore, which company is the laser source from used in this experiments?
Line 102- 103 Figure 1: The authors illustrated how they define the fracture zone or cut kerf. This seems to be rather random because the cut kerf if way larger just a little bit underneath the shown “fracture width”. How was the cut kerf evaluated? Just by measuring the minimal kerf width? What about the difference in the kerf width inside of one parameter cut?
Results:
Line 142 Table 2: It is not clear why to mention the process parameters again since they are already mentioned in table 1? There is no need of showing them again.
Line 143 Figure 2: This display is rather confusion then helping to understand the relationship between the process parameters. It would make more sense to display this in a graph using the kerf width on the Y-axis and the applied cumulate laser fluence (which include all the processing parameters (rep.rate, Pulse energy, Scanning speed)) on the X-axis and then discuss the results and trends.
Line 135-137: “While the scanning speed is higher than 600 mm/s, the instantaneous thermal effect is unable to fuse the fabric leading to a kerf width of 0, meaning that the non-woven fabric is not cut” It would be favorable to show how the cutting depth is developing during the processing by different parameters and when a fully separation is confirmed. There are several other authors how already shown a measuring strategy by light microscopy of foams or woven structures. This would improve the impact of the paper even further and not only focus on the kerf width.
Line 167 – 291: The part with the EDX is completely wrong and non-scientific. 1.) Is it not possible to give certain % values with decimals to describe the wt% content of the area by EDX. You need to perform XPS (X-ray photoelectron spectroscopy) to get sophisticated results. For EDX there are the 3 rules that need to be accomplished to give at least some insights. The specimens need to be (A) even/flat , (B) Homogenous and C (metallic) Non of those 3 specification are met here. Furthermore, the acceleration volage is choses to high with 15-20kV leading to a generation of a huge plume and receiving information from an entire volume and not on the kerf area. Second, the authors mention that the surface was covered with platinum for condition, but the numbers given of oxygen and carbon add up to 100%. What about the Pt?? The entire endeavor of using EDX was the wrong approach and needs to be reconsidered. Either using XPS, or molding and polishing the samples to be flat and coated afterwards, but still there will be the issue of the highly porous structure and the fact that the amount of carbon, which get “attracted” by the electrons to the surface from the chamber, will rise over the mearing time of the EDX. So if the measurement is longer, the carbon content will grow more and more obliterating the entire result.
Line 292 Figure 8: It is not clear what scientific experiment was conducted here and what should be seen from that figure. This is rather confusing than helping to understand the investigation.
Comments on the Quality of English Language
The quality seems to be reasonable. The content and the used "tools" are the biggest issue here.
Author Response
Please see the attachment。

Reviewer 2 Report
Comments and Suggestions for Authors
This paper investigated on the laser cutting of non-woven fabric, the topic of this paper is interesting, and the laser cutting experiments were conducted, the effect of laser parameters on the cutting quality was analyzed. The comments as following:
1. There are many different types of non-woven fabric, which type of non-woven fabric is used in this paper, and what are the related parameters about the used non-woven fabric?
2. This paper analyzed the effect of laser parameters on the width of cut edge, this is important for non-woven fabric cutting, the other important index is the width of the burns area, maybe this is also should be analyzed.
3. The conclusions in this paper should be improved to be more concise with several summaries.
Reviewer 3 Report
Comments and Suggestions for Authors
The following revision could enhance the overall quality of the paper:
1. The Abstract is not well-structured and needs revision for clarity and coherence:
The thickness of the sample and the maximum power of the laser should be presented in the abstract.
Some highlighted result should be presented in Abstract.
Some quantity of the outputs are better to be expressed in abstract.
The input parameters and their ranges should be explained in the abstract.
2. The last paragraph of the Introduction requires revision:
- Clearly highlight the novelty of the current research.
- Provide a concise summary of the paper’s key contributions in this section.
3. In the literature review it is recommended to refer to the recent study on laser cutting laser processing and related topics. For this you could use the following papers in the introduction:
Friction behavior of biodegradable electrospun polyester nanofibrous membranes. Tribology International, 188, 108891. doi: https://doi.org/10.1016/j.triboint.2023.108891
In-situ growth of robust superlubricated nano-skin on electrospun nanofibers for post-operative adhesion prevention. Nature Communications, 13(1), 5056. doi: https://doi.org/10.1038/s41467-022-32804-0
Covalently grafting polycation to bacterial cellulose for antibacterial and anti-cell adhesive wound dressings. International Journal of Biological Macromolecules, 269, 132157. doi: https://doi.org/10.1016/j.ijbiomac.2024.132157
4. Instead of Repetition rate, you need to write Frequency. Replace it in the whole paper.
5. In Table 1, present a column for sample number. And in other sections use these numbers to mention them.
6. Mention the maximum power of the used laser. And explain more about the laser type in the manuscript.
7. Important information of the study is not written well!!!
The method of measuring outputs.
Input parameters and outputs have to be presented in a clear Table. In a good format.
8. Talk about the gas pressure and it kind and effect more in the manuscript!!!
9. Where is the position of the focal plane in the experiment? Is it fixed during all experiments? Mention all these in the paper
10. How you determine the Focal Plane Position of the laser beam before experiments?
11. The conclusions are better to mentioned in separate numbered sentences instead of a paragraph. If possible, please.
Reviewer 4 Report
Comments and Suggestions for Authors
In this study, an UV nanosecond pulsed laser with wavelength of 355 nm is used to cut non-woven fabric. The variation of kerf width, surface morphology and chemical contents are investigated respectively under different laser processing parameters and the optimal processing parameter is determined. The experimental results demonstrate that the degree of crystallization and chemical composition of the kerf on the non-woven fabric surface are significantly influenced by laser cutting parameters such as laser scanning speed and repetition rate. The work done is purely experimental. The review suggests the reference is added: “Prediction of temperature-dependent transverse strength of carbon fiber reinforced polymer composites by a modified cohesive zone model”. Composite Structures, Vol 304, Part 1, 116310, 2023. This is to help the current research be extended to analytical/numerical simulations of laser cutting for non-woven fabric materials.
The manuscript has been revised based the previous reviewer’s comments. The investigations conducted in the manuscript addressed a useful engineering problem which has good application potentials for the related industries. The reviewer recognizes this contribution and recommends the publication of the manuscript.
Round 2
Reviewer 1 Report
Comments and Suggestions for Authors
Thanks for the reviewer’s valuable comments. The content of lines 42-52 discusses the development of laser cutting technology and its main applications. The purpose of this content is mainly to indicate the advantages and high processing efficiency of laser cutting technology in hard materials such as metals and ceramics. This can help to demonstrate the advantages of laser cutting for soft and stretchable materials, including high processing efficiency and cutting quality. In addition, we have added literature review for the current status of laser cutting technology for non-woven fabric in order to explain the current state of the technology with more details.
The Authors needs to point out the work that has already been done by other researchers in the field of cutting polymers as well as those fabric materials. This helps to show and identify the research gab that is still open.
Thanks for the reviewer’s valuable comments.
The spot diameter given in the diagram is the 2w value. The values of other laser processing parameters such as pulse width can be found in the manual of the laser equipment.
Please then indicate the source of the manual and where it could be found.
Thanks for the reviewer’s valuable comments.
In order to evaluate the cutting quality and processing efficiency, the minimum kerf width in the SEM image was selected. For the same set of laser processing parameters, due to the randomness and uncertainty of laser cutting process, there is a difference in the kerf width that is unavoidable. However, the main purpose of this work is to ensure that the non-woven fabric is completely cut off and the effects of the laser processing parameters are further investigated. Therefore, we believe that it can be more direct and simpler to use the smallest kerf width value as an indicator in this work. We appreciate the reviewer’s understanding.
The authors still does not fully understand what the major challenge is even when they pointed out it itself “due to the randomness and uncertainty of laser cutting process, there is a difference in the kerf width that is unavoidable”. And because of the reason the authors should know that if an uncertain process takes place you need to address that by error bars for instance or giving an average value with error. Just “looking” where the width is the smallest is the most unscientific way of working. “However, the main purpose of this work is to ensure that the non-woven fabric is completely cut off and the effects of the laser processing parameters are further investigated.” The kerf width is on of the most important processing parameter so you need to put here more effort in. Currently it is no scientific at all.
Thanks for the reviewer’s valuable comments. We deeply appreciate the reviewer’s guidance on the analysis for the EDS results. Surface scanning function of the EDS gives the wt% value of the chemical elements in the scanned area, which can help to analyze the surface chemistry of the laser cut non-woven fabric with good details. As a matter of fact, we referred to the following work (Tsai H Y, Yang C C, Hsiao W T, Huang K C, Yeh J A, Analysis of fabric materials cut using ultraviolet laser ablation, Appl. Phys. A-Mater. Sci. Process., 122 (2016) 304.), in which EDS has also been used to analyze the surface chemistry of laser cut fabric materials. Therefore, we believe that EDS has been well enough for the surface chemistry analysis in this work. However, we also agree with the reviewer that XPS analysis can be very helpful, and we will consider using such analysis technique for the future work. We appreciate the reviewer for the understanding. The surface of the laser cut non-woven fabric prepared by this research is relatively flat and the surface roughness is low. In addition, before the EDS experiments, we sprayed it with platinum to enhance its electrical conductivity in order to meet the requirements of the EDS measurement. Because the kerf area is relatively narrow and elongated, the experimental results obtained using the EDS line scanning function cannot be used for analysis. Therefore, we chose the surface scanning function, although the scanning was performed on the entire non-woven fabric surface. But it is believed that the measurement results can accurately reflect the chemical elemental change on the kerf, and thus the use of surface scanning function is considered proper. The appearance of Pt element on the surface is due to the Pt spraying before the EDS tests, which does not affect the measurement of the C and O elements. For the final data processing, we removed the content of Pt element and normalized the content of C and O elements, since these two elements are the key factors for surface chemistry analysis. For highly porous structures, they are prone to be "attracted" by electrons from the chamber to the surface, resulting in a rise in carbon over the EDS measurement duration. We have tried to utilize a shorter detection time to reduce its impact. Again, we would like to appreciate the reviewer for raising the possible problems for the EDS analysis and understanding, and we will carefully analyze them in the subsequent research.
Just by pointing out the wrong methods used by others does not make your methods from wrong to right. Still, based on the answers the authors did not understand why an EDS method here cannot be used to give quantitative numbers, and even with 2 decimals. If you want to characterize the impact of your process you need to use the current methods, otherwise it no science. There is no room for understanding on my side, it is just the wrong method.
Thanks for the reviewer’s valuable comments.
Figure 8 is a schematic of the skin friction experiment. The relevant experimental details can be found with more details in lines 283-292. As can be seen from the figure, after the friction experiment with the laser cut non-woven fabric, the arms of the two experimenters did not show obvious redness and swelling. The non-woven fabric kerf after laser cutting on the surface is smooth and will not cause harm to human skin.
This is a single test, with the amount of test specimen on two!! For making a point the authors needs to make many more test on different specimen and proof QUANTITIVLY the impact of this study. This picture does not show anything at all and does not give any useful information. There is no measurement whatsoever.
Reviewer 2 Report
Comments and Suggestions for Authors
All comments have been revised.
Reviewer 3 Report
Comments and Suggestions for Authors
Following a professional revision by the authors, I can now recommend it for publication, as the requested corrections have been made precisely and discussed thoroughly.